# Antitumor Efficacy of Doxorubicin-Loaded Electrospun Attapulgite–Poly(lactic-co-glycolic acid) Composite Nanofibers

**DOI:** 10.3390/jfb13020055

**Published:** 2022-05-10

**Authors:** Zhe Wang, Yili Zhao, Mingwu Shen, Helena Tomás, Benqing Zhou, Xiangyang Shi

**Affiliations:** 1Department of Biomedical Engineering, College of Engineering, Shantou University, Shantou 515063, China; wangzhe20090716@126.com; 2Shanghai Engineering Research Center of Nano-Biomaterials and Regenerative Medicine, College of Chemistry, Chemical Engineering and Biotechnology, Donghua University, Shanghai 201620, China; mwshen@dhu.edu.cn; 3College of Textile Science and Engineering, Zhejiang Sci-Tech University, Hangzhou 310018, China; yzhao@zstu.edu.cn; 4CQM—Centro de Química da Madeira, Universidade da Madeira, Campus da Penteada, 9000-390 Funchal, Portugal; lenat@staff.uma.pt

**Keywords:** nanofibers, attapulgite, drug release, poly(lactic-co-glycolic acid), antitumor therapy

## Abstract

Currently, cancer chemotherapeutic drugs still have the defects of high toxicity and low bioavailability, so it is critical to design novel drug release systems for cancer chemotherapy. Here, we report a method to fabricate electrospun drug-loaded organic/inorganic hybrid nanofibrous system for antitumor therapy applications. In this work, rod-like attapulgite (ATT) was utilized to load a model anticancer drug doxorubicin (DOX), and mixed with poly(lactic-co-glycolic acid) (PLGA) to form electrospun hybrid nanofibers. The ATT/DOX/PLGA composite nanofibers were characterized through various techniques. It is feasible to load DOX onto ATT surfaces, and the ATT/DOX/PLGA nanofibers show a smooth and uniform morphology with improved mechanical durability. Under neutral and acidic pH conditions, the loaded DOX was released from ATT/DOX/PLGA nanofibers in a sustained manner. In addition, the released DOX from the nanofibers could significantly inhibit the growth of tumor cells. Owing to the significantly reduced burst release profile and increased mechanical durability of the ATT/DOX/PLGA nanofibers, the designed organic–inorganic hybrid nanofibers may hold great promise as a nanoplatform to encapsulate different drugs for enhanced local tumor therapy applications.

## 1. Introduction

Electrospinning is a straightforward and universal nanomanufacturing technology for producing ultralong nanofibers with adjustable diameters ranging from several nanometers to microns [1,2,3,4,5]. Different types of polymers including poly(lactic-co-glycolic acid) (PLGA) [6,7,8], polycaprolactone [9,10], polyurethane [11], polyvinyl pyrrolidone [12,13], and polyvinyl acetate [14,15] have been electrospun to form nanofibers. The electrospun nanofibers show various applications, including but not limited to their use in sensors [16,17,18], supercapacitors [19,20,21,22], dye-sensitized solar cells [23,24,25], wound dressing [26,27,28], tissue engineering [29,30], and drug delivery systems [31,32,33,34,35].

Conventional [36,37,38], coaxial [39,40,41], and emulsion [42,43,44] electrospinning are commonly used techniques to prepare nanofiber-based drug delivery systems. The traditional single-fluid electrospinning method can directly integrate drug molecules into nanofibers through simple electrospinning of mixed solution of drug and polymer, or adsorption of the drug on or within pre-formed nanofibers [36,45]. Coaxial and emulsion electrospinning, which both achieve sustained drug release, can be regarded as improved methods for drug delivery systems to alleviate burst drug release [42,46]. In these methods, with drug molecules being incorporated into the core area of nanofibers, a “core-shell” structure was constructed, where the outer polymer shell could control drug release like a barrier [47]. However, the emulsifier used for emulsion electrospinning often shows cytotoxicity, and the coaxial electrospinning usually requires higher electrospinning parameters. Additionally, conventional electrospun nanofibers consisting of polymers frequently face issues of mechanical durability that make them unsuitable for practical biomedical applications. Hence, it is essential to develop novel nanofiber systems that can prevent drugs from burst release and have high mechanical durability.

Our previous work has indicated that drug-loaded halloysite nanotubes (HNTs) [48,49], laponite (LAP) [50], and nano-hydroxyapatite (n-HA) [51,52] can be combined with PLGA nanofibers through a simple electrospinning method. The formed hybrid nanofibers displayed a decreased burst drug release manner and enhanced mechanical durability. Here, the HNTs, LAP, and n-HA themselves served as nanocarriers for drugs, and could effectively enhance the mechanical durability of the nanofibers. This previous work inspired us to incorporate other inorganic nanoparticles with the ability to loading drug molecules into polymer nanofibers, for mitigating the burst release of drugs and improving the mechanical durability of the nanofibers.

Attapulgite (ATT) is a type of naturally mined clay that is cost-effective and shows large specific surface, high porosity, and high surface activity as well as good biocompatibility [5,53,54]. Due to the high porosity of ATT, its small dimensions, high aspect ratio and relatively uniform morphology, ATT-doped nanofibers showed a much higher drug loading efficiency than the reported HNT-, LAP-, and n-HA-based electrospun nanofibers, which is beneficial for antitumor applications [49,50,51]. In addition, ATT could significantly enhance the mechanical durability of polymer materials [55,56,57]. However, weak interactions between drug molecules such as doxorubicin (DOX) and ATT result in sudden release of drug molecules from ATT/drug nanocomplexes. Therefore, it is quite reasonable to design an ATT-doped polymer nanofiber system, where polymer nanofibers and ATT both serve as satisfactory vehicles to load drug molecules, enabling drugs to show sustained release profiles. Furthermore, the ATT-doped polymer nanofibers are expected to have improved mechanical durability.

Here, DOX-loaded ATT was used to dope PLGA nanofibers to slow down the speed of drug release and enhance the mechanical durability of the nanofibers for enhanced local tumor therapy applications. Firstly, DOX was loaded on the ATT surface through physical adsorption, followed by mixing with PLGA solution for the preparation of electrospun nanofibers. Various analysis techniques were utilized to characterize the ATT/DOX/PLGA nanofibers. The DOX release manner of the ATT/DOX/PLGA nanofibers was evaluated by UV-vis spectroscopy under different pH conditions, and the antiproliferative activity was evaluated by resazurin reduction assay and phase contrast microscopic morphology observation. To our knowledge, this is the first report on the development of ATT-doped PLGA nanofibers with desirable drug release speed and mechanical durability for *in vitro* anticancer treatment applications.

## 2. Materials and Methods

### 2.1. Preparation of ATT/DOX/PLGA Nanofibers

ATT water solution with different concentrations (5, 7.5, 10, 12.5, 15, or 17.5 mg/mL) and DOX water solution (2 mg/mL) were prepared at room temperature. Then equal amounts of ATT and DOX solutions were fully mixed and magnetically stirred in the dark at room temperature for 24 h. The mixture was centrifugated for 5 min at 5000 rpm and washed by water to produce the ATT/DOX complexes.

The formed ATT/DOX complexes were then mixed with PLGA solution before electrospinning. For comparison, an equal amount of ATT (4 wt% to PLGA) or DOX (0.5 wt% to PLGA) was added to PLGA solution and magnetically stirred for 2 h before electrospinning. The procedure of formulation of electrospun nanofibers was similar to our previous reports [49,52]. The formed electrospun nanofibers were vacuum-dried at room temperature for more than 48 h to completely evaporate the residual organic solvent.

### 2.2. In Vitro Antiproliferative Activity Evaluation

A certain quantity of experimental and control samples was weighed, and all samples underwent UV sterilization overnight. All the samples were placed in a 6-well cell culture plate, and about 2 mL of 75% alcohol was added to soak and disinfect for 5 min, while UV irradiation was carried out for sterilization. Then, the alcohol was sucked out and each well was washed with PBS three times. We then added an appropriate volume of RPMI 1640 medium into each well (the theoretical concentration of DOX in nanofibers was 200 μg/mL), and placed the cell culture plates in a CO_2_ incubator for 24 h at 37 °C. The release medium of each well was collected for evaluation of *in vitro* antiproliferative activity.

Cell suspension with a density of 1 × 10^5^ cells/mL was prepared, and 20 μL was added to a 96-well plate, so that the plating density of CAL 72 cells was 2 × 10^3^ cells/well. Samples used for testing were: 100, 50, 20, or 10 μL fresh medium, free DOX solution ([DOX] = 200 μg/mL), ATT/DOX solution ([DOX] = 200 μg/mL), ATT particle solution with equivalent concentration of the ATT/DOX complexes, and immersion medium of the nanofibers of DOX/PLGA ([DOX] = 200 μg/mL according to theoretic DOX loading within the fibers), PLGA with equivalent DOX concentration of the release medium of DOX/PLGA, ATT/PLGA with equivalent DOX concentration of the release medium of DOX/PLGA, and ATT/DOX/PLGA ([DOX] = 200 μg/mL according to theoretic DOX loading within the fibers). The total volume was added to 200 μL/well using fresh medium. Each sample and concentration gradient were stored six parallel, and the samples were shaken well and then cultured in an incubator. The initial concentration of DOX is 200 μg/mL, so the concentration gradients after dilution were 100, 50, 20 and 10 μg/mL, respectively. After one day of culture at 37 °C and 5% CO_2_, the original medium was discarded, and 180 μL fresh medium and 20 μL resazurin solution added into each well (0.1 mg/mL dissolved in PBS and filtered for sterilization). Four hours after re-culture, cell viability was detected by a microplate reader (model Victor3 1420, PerkinElmer, Waltham, MA, USA). Meanwhile, the morphology of the treated cancer cells was observed by an inverted optical microscope (Nikon Eclipse TE 2000E, Tokyo, Japan).

## 3. Results and Discussion

### 3.1. Synthesis and Characterization of ATT/DOX Complexes

The rod-like ATT was utilized to load DOX via physical adsorption. The DOX loading efficiency was optimized by regulating the ATT and DOX feed mass ratios. As shown in Appendix A, the DOX loading efficiency increased synchronously with the ATT/DOX mass ratio. At the ATT/DOX mass ratio of 7.5: 1, DOX loading efficiency was up to 97.2%. We selected this feed ratio to prepare ATT/DOX complexes and ATT/DOX/PLGA nanofibers. Fourier Transform Infrared (FTIR) spectroscopy (GMI, Ramsey, MN, USA) was utilized to qualitatively analyze the ATT/DOX complexes (Figure 1). For free ATT, as shown in Figure 1a, the absorption peaks at 3546 and 3434 cm^–1^ were the O-H stretching vibration of water in ATT. The absorption band at 1654 cm^–1^ was the bending vibration of absorbed and zeolitic water in channels [58], and the bands at 1197 and 985 cm^–1^ were the fingerprints of fibrous clay minerals (e.g., ATT) [59,60]. For the ATT/DOX complexes, the typical absorption peaks at 3549 and 3436 cm^−1^ were assigned to the bending vibrations of the N-H of loaded DOX, and the peaks at 1414 cm^−1^ might be attributed to the in-plane stretching vibrations of the C–C single bond skeleton of DOX (Figure 1a). Compared with the spectrum of free ATT, new peaks emerged at 1414, 1029, 802, 684 cm^−1^ in the spectrum of the ATT/DOX complexes, which were probably due to the combination of the peaks of free DOX at 1621, 1413,1073, and 1001 cm^−1^ (Figure 1b) and ATT at 1654,1198, 792, 649 cm^−1^ (Figure 1a). These results indicated that DOX was loaded onto ATT surfaces successfully.

X-ray diffraction (XRD) was used to observe the crystalline structure of ATT before and after DOX loading (Figure 2). The characteristic diffraction peaks at 2θ of 8.34°, 13.70°, 19.88°, 27.64°, and 34.32° were assigned to (110), (200), (040), (400) and (102) crystal planes of ATT, respectively, according to the literature [61,62]. This indicates that the loading of DOX did not significantly change the crystal structure of ATT. The analysis of diffraction angle and plane spacing was also performed using XRD (Appendix A). The loading of DOX caused the 2θ peaks in the (200) plane to vary from 13.70° to 13.58°, and in the (400) plane to vary from 27.64° to 27.58°. In addition, the plane spacing of (200) increased slightly from 6.459 to 6.515 A, indicating that the adsorption of DOX on the ATT surface occurs preferentially in the plane of (200). It is noteworthy that the XRD patterns of ATT/DOX showed no peaks related to DOX, probably because of the amorphous state of the loaded DOX.

The loading of DOX onto the ATT surfaces was also studied by UV−vis spectroscopy. As shown in Appendix A, free DOX and the ATT/DOX showed a DOX-related characteristic absorption peak at around 490 nm, while free ATT did not show any associated peak. This further demonstrated that DOX could be successfully loaded by ATT.

### 3.2. Construction and Characterization of ATT/DOX/PLGA Nanofibers

The DOX-loaded ATT was incorporated within PLGA nanofibers by electrospinning to form the ATT/DOX/PLGA composite nanofibers. The ATT/DOX/PLGA nanofibers were characterized by TEM and fluorescence microscopic imaging. TEM images of the ATT/DOX complexes showed that the DOX loading onto the surface of the ATT had no obvious impact on the morphology of the ATT, as compared to free ATT [63] (Appendix A). The shapes of rod-like ATT/DOX complexes and PLGA nanofibers could be clearly observed on TEM images (Figure 3a,b). The ATT/DOX complexes could also be seen in the ATT/DOX/PLGA nanofibers (Figure 3c). This indicated that the ATT/DOX complexes could be effectively incorporated within PLGA nanofibers by electrospinning. As shown in Appendix A, the entire fiber of the DOX/PLGA showed DOX-related red fluorescence, which was similar to the fluorescence microscope image of the ATT/DOX/PLGA composite nanofibers (Figure 3d). Therefore, the release of DOX from ATT could occur prior to the electrospinning during 2 h stirring of the PLGA/ATT/DOX solution. These results further confirmed that DOX could be effectively loaded by the fibers.

Similar to our previous studies [49,52], the ATT/DOX/PLGA nanofibers displayed a smooth and uniform fibrous morphology, as good as free PLGA and DOX/PLGA nanofibers (Figure 4). The diameter of the ATT/DOX/PLGA nanofibers was 656 nm, smaller than those of free PLGA (756 nm) and DOX/PLGA (728 nm) nanofibers, which was possibly related to the increase of solution conductivity by the introduction of ions (Figure 4). Additionally, with the increase of ATT content, both the diameter and porosity of the nanofiber decreased. As shown in Appendix A, the porosity of the nanofiber mat decreased to 73.8% with the addition of DOX and ATT, smaller than that of the free PLGA fiber mat (84%). The reason may be the reduction of fiber diameter after incorporation of ATT and DOX.

The surface hydrophilicity of nanofibers is critical for their interaction with cells. We studied the effects of ATT and ATT/DOX incorporation on the surface hydrophilicity of PLGA nanofibers (Appendix A and Appendix A). The contact angle of PLGA nanofibers (136.8 ± 2.6°) decreased to 131.9 ± 2.7° (ATT/PLGA) and 129.1 ± 1.5° (ATT/DOX/PLGA), respectively after incorporation with ATT and ATT/DOX complexes. This indicated that the hydrophilicity of PLGA nanofiber was slightly improved by ATT incorporation, which may facilitate the infiltration of nutrients to promote the adhesion and proliferation of cells.

Notably, the mechanical properties of PLGA nanofibers were improved after doping ATT or ATT/ DOX complexes. It was obvious that the breaking strength and Young’s modulus of the ATT/PLGA and ATT/DOX/PLGA nanofibers were much higher than those of PLGA nanofibers (Figure 5), which was also illustrated by the quantitative data shown in Appendix A. The failure strains of ATT/PLGA and ATT/DOX/PLGA nanofibers were slightly lower than that of PLGA nanofibers, which might be caused by the increased brittleness of the nanofibers after incorporation with ATT or ATT/DOX complexes.

### 3.3. In Vitro Drug Release

For antitumor therapeutic applications, it is important to understand the release kinetics of a drug delivery system. We then explored the release kinetics of DOX from the ATT/DOX/PLGA nanofibers in different pH conditions. For comparison, DOX release from the ATT/DOX complexes and DOX/PLGA nanofibers was also investigated. As shown in Figure 6, under pH conditions of 5.5 and 7.4, DOX from ATT/DOX complexes and DOX/PLGA nanofibers had much higher release rates compared to that from ATT/DOX/PLGA nanofibers. The DOX release of ATT/DOX/PLGA nanofibers under both pH conditions showed a sustained release profile, with approximately 10% of DOX release amounts within 10 days. It should be noted that the DOX release amount from ATT/DOX/PLGA nanofibers in the first 24 h at pH 5.5 was about 10%, while at pH 7.4 was about 5%. In addition, under pH 7.4 conditions, the DOX release amounts from the DOX/PLGA fibers in the first 8 h and ATT/DOX in the first 24 h were about 34.4% and 21.2%, respectively. Meanwhile, under pH 5.5, the DOX releases from the DOX/PLGA fibers and ATT/DOX complexes in the first 24 h were 63.8% and 45.0%, respectively. Therefore, DOX release from DOX/PLGA fibers, ATT/DOX complexes, and ATT/DOX/PLGA nanofibers all showed a pH response drug release. The reason for the sustained release of DOX from the ATT/DOX/PLGA composite nanofibers should be that DOX needs to be released from ATT first and then released from the PLGA matrix, thus avoiding burst release effect.

### 3.4. In Vitro Antiproliferative Efficacy

In order to develop an efficient drug delivery system, it is necessary to retain the antitumor activity of drug molecules encapsulated in the ATT/PLGA nanofibers. Therefore, we used DOX release medium of the ATT/DOX/PLGA nanofibers to evaluate *in vitro* antitumor efficacy of the nanofibers. As shown in Figure 7, ATT, the released media from PLGA fibers, and ATT/PLGA fibers without DOX loading did not display any significant cytotoxicity at different concentrations. In contrast, free DOX, and the DOX release medium of the ATT/DOX and DOX/PLGA nanofibers could inhibit the growth of CAL72 cells efficiently at the given DOX concentrations. It should be noted that the DOX release medium of ATT/DOX/PLGA nanofibers exhibited obvious cytotoxicity only at DOX concentrations higher than 20 μg/mL. Under the same DOX concentrations, the viability of the cells treated by the release medium of ATT/DOX/PLGA nanofibers was a little lower than that of free DOX, and of ATT/DOX (or DOX/PLGA nanofibers). Therefore, the slow release of DOX from the ATT/DOX/PLGA nanofibers could help to reduce the dose and side effects at high initial concentrations.

To further confirm *in vitro* antiproliferative activity of the drug-loaded nanofibers, the morphologies of cells treated with free DOX, or the release medium of ATT/DOX complexes, ATT/DOX/PLGA nanofibers, or DOX/PLGA nanofibers were observed by phase-contrast microscopy for 24 h (Figure 8). They all had a large number of rounded and disconnected cells, indicating that a large percentage of the cells had died. In addition, the proportion of rounded dead cells increased with the increase in the concentration of DOX. These results were consistent with the resazurin reduction assay data.

## 4. Conclusions

In summary, we have prepared ATT/DOX/PLGA composite nanofibers through electrospinning for *in vitro* cancer therapy applications. The anticancer drug DOX can be successfully loaded onto the ATT particle surface, and the ATT/DOX/PLGA composite nanofibers have a smooth and uniform morphology with improved mechanical durability. Under neutral and acidic pH conditions, the loaded DOX can be released from ATT/DOX/PLGA nanofibers in a sustained manner. Furthermore, the released DOX from the nanofibers could significantly inhibit the growth of cancer cells *in vitro*. The ATT/DOX-doped PLGA nanofibers can significantly improve the mechanical properties of electrospun PLGA nanofibers, which could be used as a functional material for tissue engineering scaffold for enhanced local tumor therapy applications. In addition, ATT particles and PLGA nanofibers have dual sustained-release effects on drug molecules. In future work, the dispersion of ATT and DOX/ATT in electrospinning solution may be further improved through the surface modification of ATT, so that ATT can maximize the mechanical properties of the fibers. Further *in vivo* experiments should also be performed to prove the antitumor efficacy of these composite nanofibers.

## Figures and Tables

**Figure 1 jfb-13-00055-f001:**
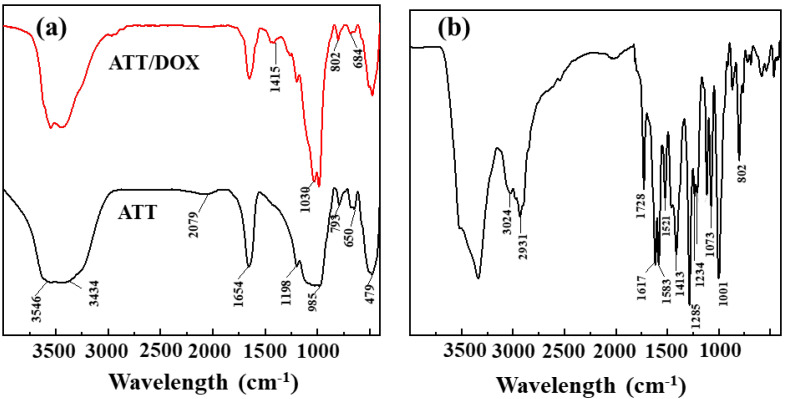
FTIR spectra of ATT and ATT/DOX complexes (**a**), and free DOX (**b**).

**Figure 2 jfb-13-00055-f002:**
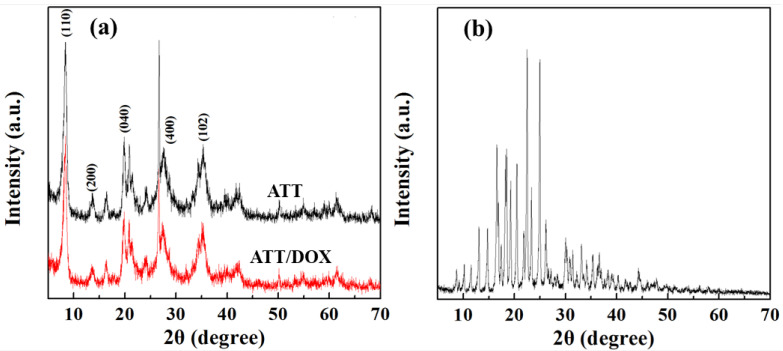
The XRD crystalline structure of ATT and ATT/DOX complexes (**a**), and free DOX (**b**).

**Figure 3 jfb-13-00055-f003:**
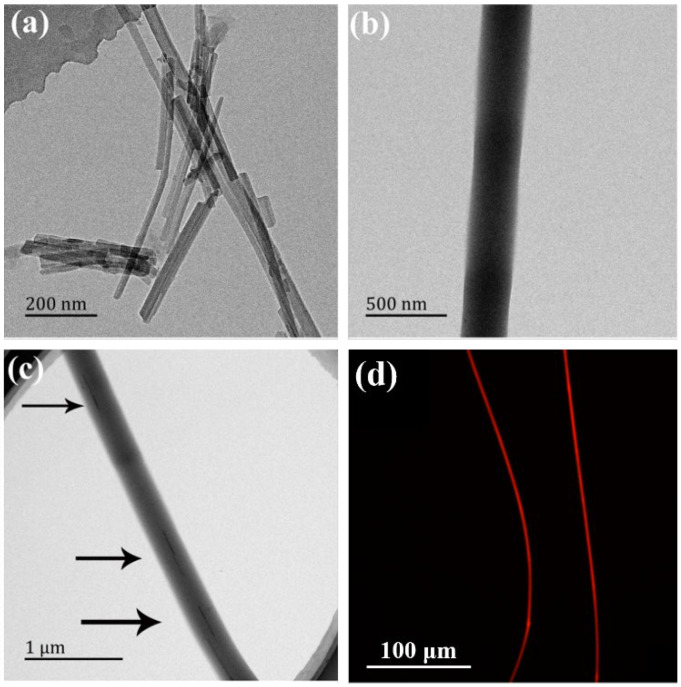
Transmission electron microscopy (TEM) images of ATT/DOX complexes (**a**), PLGA nanofibers (**b**), ATT/DOX/PLGA composite nanofibers (**c**). The arrows point to the ATT/DOX complexes. (**d**) A fluorescence microscopic image of the ATT/DOX/PLGA composite nanofibers (DOX has a maximum excitation and emission wavelength of 470 and 560 nm, respectively).

**Figure 4 jfb-13-00055-f004:**
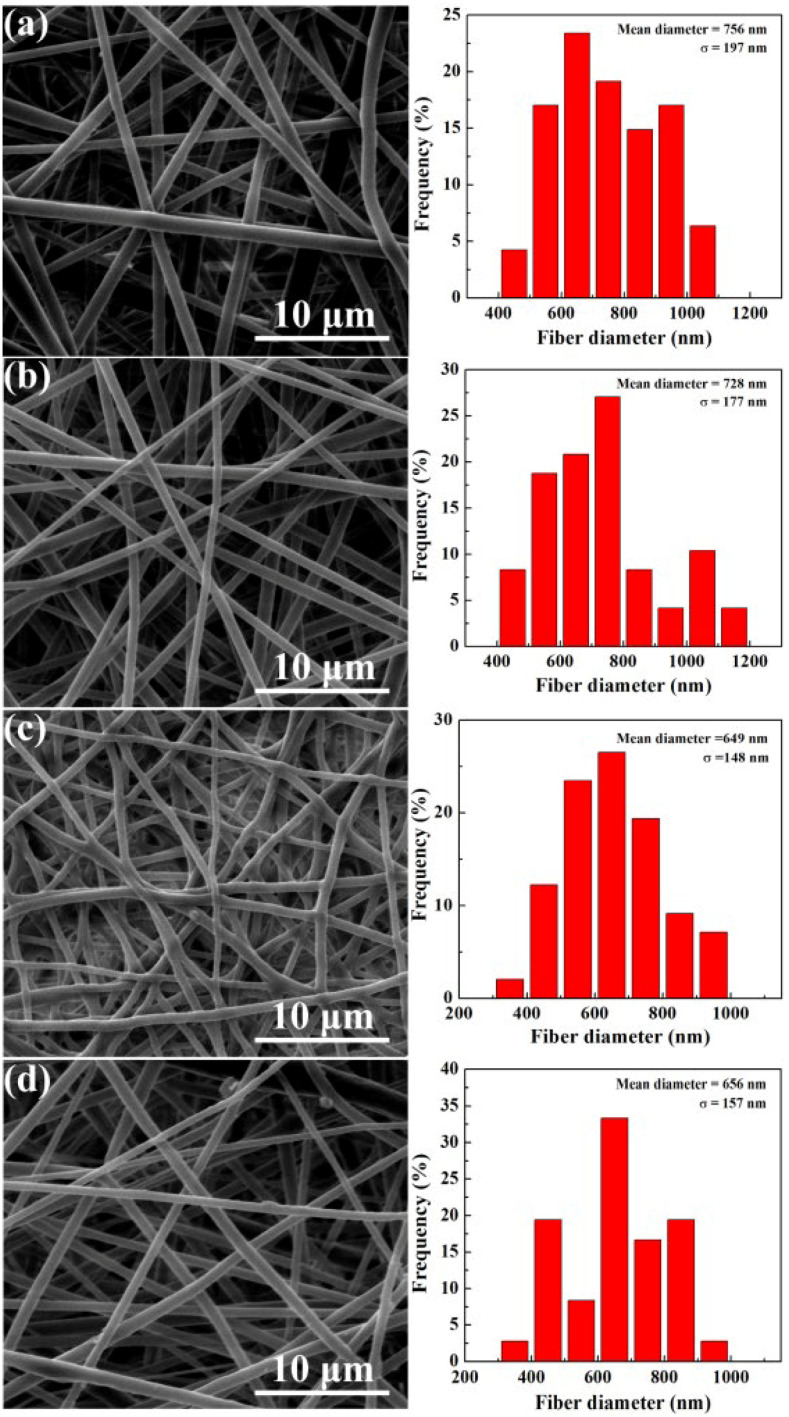
SEM images and diameter distribution histograms of free PLGA (**a**), DOX/PLGA (**b**), ATT/PLGA (**c**), and ATT/DOX/PLGA (**d**) nanofibers.

**Figure 5 jfb-13-00055-f005:**
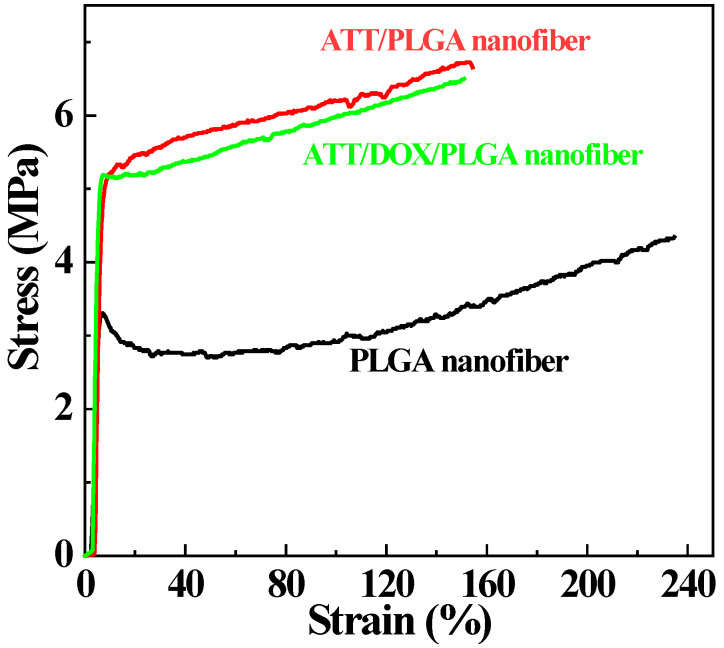
The strain–stress curves of PLGA, ATT/PLGA, and ATT/DOX/PLGA nanofibers, respectively.

**Figure 6 jfb-13-00055-f006:**
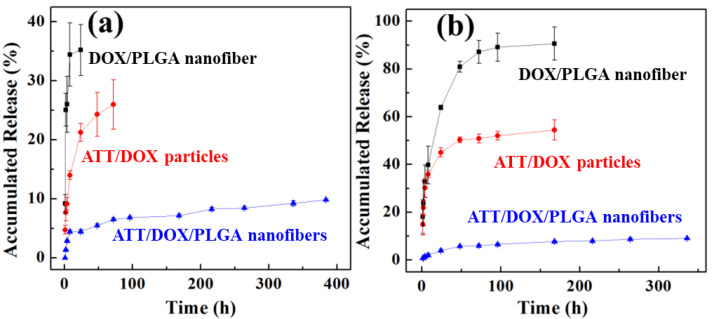
*In vitro* release of DOX from ATT/DOX complexes, DOX/PLGA and ATT/DOX/PLGA nanofibers at pH 7.4 (**a**) and 5.5 (**b**), at 37 °C.

**Figure 7 jfb-13-00055-f007:**
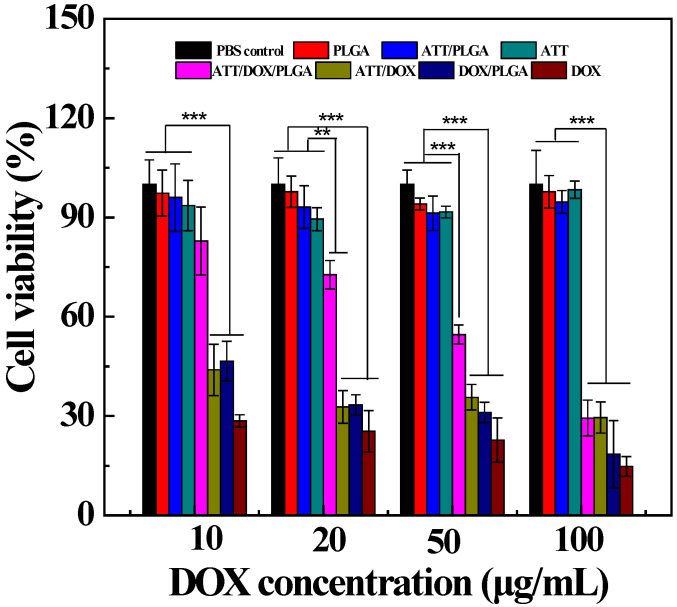
Resazurin reduction assay of CAL72 cells treated with free DOX, and the release medium of ATT/DOX complexes, DOX/PLGA and ATT/DOX/PLGA nanofibers at different DOX concentrations for 24 h (** *p* < 0.01 and *** *p* < 0.001). The PLGA nanofibers, ATT particles, and ATT/PLGA nanofibers without DOX loading were also utilized for cell treatment cells under similar conditions.

**Figure 8 jfb-13-00055-f008:**
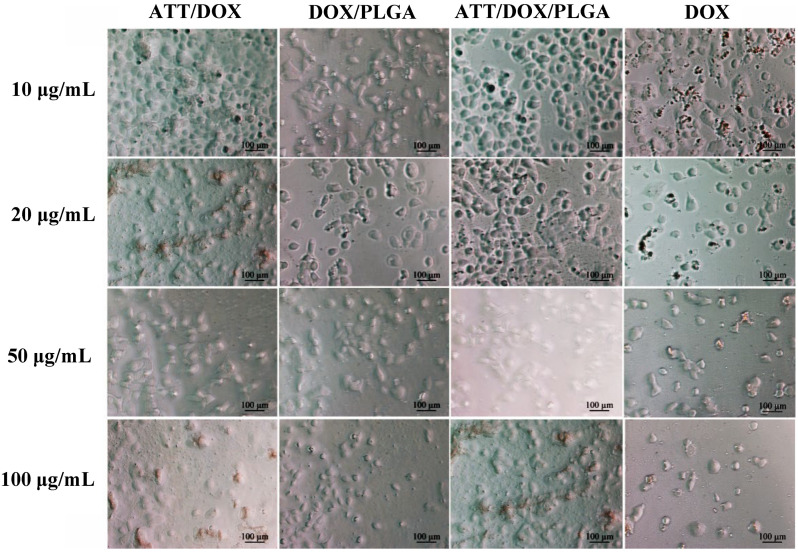
The micrographs of CAL72 cells treated with free DOX, and the release media of ATT/DOX complexes, DOX/PLGA and ATT/DOX/PLGA nanofibers with different DOX concentrations for 24 h.

## Data Availability

The data presented are available from the corresponding author upon request.

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
