# Peer review of "Antitumor Efficacy of Doxorubicin-Loaded Electrospun Attapulgite–Poly(lactic-co-glycolic acid) Composite Nanofibers"

_jfb, 2022, doi:10.3390/jfb13020055_

Round 1

Reviewer 1 Report

This work details the fabrication of electrospun attapulgite (ATT)/doxorubicin/PLGA nanofibers for in vitro anti-cancer studies. However, there are multiple queries and limitations which the authors need to address, and probably multiple experiments are required to support the publication of this article.

  • Rationale of using ATT is not clear in this work. Although authors claim ATT can improve the adsorption of DOX because of their higher surface area, there is no clear advantage of using this material compared to previously used materials.
  • In figure 3, authors show adsorption of ATT on the surface. If the DOX was adsorbed only on ATT, there should be red fluorescence only at the ATT adsorbed areas. However, entire fiber is being showing red fluorescence. Does this indicate the release of DOX from ATT prior to the electrospinning during 2h stirring of PLGA/ATT/DOX solutions? Therefore, please compare the fluorescence microscope image with DOX/PLGA nanofibers
  • In the favor of cancer microenvironment, it is beneficial to see enhanced drug release in the acidic pH conditions over the neutral pH. Although there is a definite sustained release, it doesn’t serve the purpose of pH responsive drug release that is generally required for cancer therapy.
  • The materials and methods are very poorly written and no information regarding any characterization techniques, drug release or in vitro toxicity studies. How should one understand how the experiments were performed?
  • It is important to know what the incubation conditions and after how long the release media from the electrospun nanofibers were incubated with cells. It would be worth to study time dependent release media mediated toxicity studies.
  • I believe In vivo experiments should be performed to prove the efficacy of the developed composite nanofibers.

Author Response

Reviewer #1:

Comments to the Author:

This work details the fabrication of electrospun attapulgite (ATT)/doxorubicin/PLGA nanofibers for in vitro anti-cancer studies. However, there are multiple queries and limitations which the authors need to address, and probably multiple experiments are required to support the publication of this article.

Comment 1:

Rationale of using ATT is not clear in this work. Although authors claim ATT can improve the adsorption of DOX because of their higher surface area, there is no clear advantage of using this material compared to previously used materials.

Author reply: We thank the reviewer for his/her great comments. According to the reviewer’s suggestion, we have added the description of the advantages of using ATT for the composite nanofibers. ATT as one type of naturally mined clay is cost-effective and shows large specific surface area, high porosity, and high surface activity beyond good biocompatibility, comparable to other materials, such as hydroxyapatite nanoparticles, graphene, laponite, and mesoporous silica, etc. In addition, ATT could significantly enhance the mechanical durability of polymer materials (Polymer, 2007, 48(9), 2720-2728; Polymer Degradation and Stability 2010, 95, 1581-1587; and Carbohydrate Polymers 2007, 68, 367-374). Therefore, ATT could be used as an effective vehicle to load drug molecules and doped within polymer nanofibers to increase their mechanical properties. To make it clearer, we have added two sentences on Page 4 Line 4-6 in the revised manuscript. See below:

“Attapulgite (ATT) as one type of naturally mined clay is cost-effective and shows large specific surface area, high porosity, and high surface activity beyond good biocompatibility [5,53,54]. In addition, ATT could significantly enhance the mechanical durability of polymer materials [55-57].”

Comment 2:

In figure 3, authors show adsorption of ATT on the surface. If the DOX was adsorbed only on ATT, there should be red fluorescence only at the ATT adsorbed areas. However, entire fiber is being showing red fluorescence. Does this indicate the release of DOX from ATT prior to the electrospinning during 2h stirring of PLGA/ATT/DOX solutions? Therefore, please compare the fluorescence microscope image with DOX/PLGA nanofibers?

Author reply: According to the reviewer’s suggestion, we have provided the fluorescence microscope image of the DOX/PLGA composite nanofibers in Figure S3 in the revised Supporting Information. As shown in Figure S3, the entire fiber of the DOX/PLGA also showed red fluorescence, which was similar to the fluorescence microscope image of ATT/DOX/PLGA composite nanofibers. Therefore, we agree with your point that the release of DOX from ATT could occur prior to the electrospinning during 2 h stirring of PLGA/ATT/DOX solution. We have added the description on Page 8 Line 14-18 in the revised manuscript. See also below:

As shown in Figure S3, the entire fiber of the DOX/PLGA showed DOX-related red fluorescence, which was similar to the fluorescence microscope image of ATT/DOX/PLGA composite nanofibers (Figure 3d). Therefore, the release of DOX from ATT could occur prior to the electrospinning during 2 h stirring of PLGA/ATT/DOX solution.

Comment 3:

In the favor of cancer microenvironment, it is beneficial to see enhanced drug release in the acidic pH conditions over the neutral pH. Although there is a definite sustained release, it doesn’t serve the purpose of pH responsive drug release that is generally required for cancer therapy.

Author reply: As shown in Figure 6, the DOX release amount from the DOX/PLGA fibers and ATT/DOX under pH 5.5 was much higher than that under pH 7.4 at the same DOX concentrations. In addition, the DOX release amount from the ATT/DOX/PLGA nanofibers in the first 24 h under pH 5.5 was about 10%, while under pH 7.4 was just about 5%. Therefore, the DOX release from DOX/PLGA fibers, ATT/DOX complexes, and  ATT/DOX/PLGA nanofibers all showed a pH-responsive drug release. To make it clearer, we have added some descriptions on Page 11 Line 17-19 and Page 12 Line 1-2, respectively. See below:

Page 11 Line 17-19:

“It should be noted that the DOX release amount from ATT/DOX/PLGA nanofibers in the first 24 h under pH 5.5 was about 10%, while under pH 7.4 was just about 5%.”

Page 12 Line 1-2:

“Therefore, the DOX release from DOX/PLGA fibers, ATT/DOX complexes, and  ATT/DOX/PLGA nanofibers all showed a pH-responsive drug release.”

Comment 4:

The materials and methods are very poorly written and no information regarding any characterization techniques, drug release or in vitro toxicity studies. How should one understand how the experiments were performed?

Author reply: We have provided the materials and methods including materials, in vitro drug release, and characterization techniques in Supporting Information, and in vitro antiproliferative activity evaluation in the revised manuscript. See below:

Materials

PLGA (molecular weight = 81000 g/mol) with a lactic acid/glycolic acid ratio of 50: 50 was purchased from Jinan Daigang Biotechnology Co., Ltd. (Jinan, China). DOX was purchased from Beijing Huafeng Pharmaceutical Co., Ltd. (Beijing, China). ATT was from Mingguang Jianxi Dongfeng Mine Products Factory (Mingguang, China). Dimethylsulfoxide (DMSO), tetrahydrofuran (THF), N,N-dimethyl formamide (DMF), RPMI 1640 medium, fetal bovine serum (FBS), phosphate buffer saline (PBS), penicillin, and streptomycin were purchased from Gibco (Carlsbad, CA). Resazurin was from Sigma-Aldrich (St Louis, MO). A human osteosarcoma cell line (CAL72) was from University of Madeira (Funchal, Portugal). Water used in all experiments was purified using a Milli-Q Plus 185 water purification system (Millipore, Bedford, MA) with a resistivity higher than 18 Mcm.

In vitro drug release

The release kinetics of DOX was determined by measuring the DOX absorbance at 490 nm using a UV-vis spectrophotometer. The ATT/DOX complex (5 mg) was dispersed in 2 mL of phosphate buffer saline (PBS, pH = 7.4) or sodium acetate–acetic acid buffer solution (pH = 5.4). The dispersed solution was then transferred to a dialysis tube, which was placed in a vial containing 8 mL of the corresponding buffer solution. Similarly, PLGA/DOX and ATT/DOX/PLGA nanofibers with the same DOX concentration were directly placed into different vials containing 10 mL of the corresponding buffer solution. All the samples were incubated in a vapor-bathing constant temperature vibrator with a shaking speed of 90 rpm at 37 oC for a period of 10 days. At each predetermined time point, 1 mL of outerphase solution was removed from each vial for quantitative analysis using UV-vis spectroscopy. An equal volume of fresh corresponding buffer solution was added to the vial.

Characterization techniques

The ATT/DOX/PLGA composite nanofibers were characterized via different techniques. Fourier transform infrared spectroscopy (FTIR) was performed using a Nicolet Nexus 670 FTIR spectrometer over a wavenumber range of 500 to 4000 cm-1 to confirm the loading of DOX onto the ATT particles. The crystalline structures of ATT before and after modifications were analyzed by a Rigaku D/max-2550 PC X-ray diffraction (XRD) system (Rigaku Co., Tokyo, Japan) with a wavelength of 0.154 nm at 40 kV and 200 mA. The scan was performed from 5° to 70°. Lastly, the DOX, ATT, and ATT/DOX solutions were characterized via UV-vis spectroscopy (Perkin Elmer Lambda 25, Waltham, MA) at a wavelength range of 200-900 nm. The morphology of the ATT/DOX nanohybrid was characterized by transmission electron microscopy (TEM, JZM-2100, Japan) at an operating voltage of 200 kV. The morphologies of PLGA, ATT/PLGA, PLGA/DOX, and ATT/DOX/PLGA nanofibers were also observed by SEM (JEOL JSM-5600LV, Tokyo, Japan) at a voltage of 15 kV. The porosity, mechanical properties, and the surface hydrophilicity of the nanofibers were measured according to protocols described in our previous work [1].”

2.2. In vitro antiproliferative activity evaluation

A certain amount of experimental and control samples was weighed, and all samples needed UV sterilization overnight. All the samples were placed in a 6-well cell culture plate, and then about 2 mL of 75% alcohol was added to soak and disinfect for 5 min, while UV irradiation was carried out for sterilization. Then, the alcohol was sucked out and each well was washed repeatedly with PBS for 3 times. We then added an appropriate volume of RPMI 1640 medium into each well (the theoretical concentration of DOX in nanofibers was 200 μg/mL), and put the cell culture plate in a CO2 incubator for 24 h at 37 oC. The immersion medium of each well was collected for subsequent CAL72 cell culture.

CAL72 cells (2 × 103 cells per well) were plated into a 96-well plate and cultured in a humidified incubator overnight. Then, the medium was replaced with fresh medium containing free DOX, and the release medium of the ATT/DOX complex, DOX/PLGA or ATT/DOX/PLGA nanofibrous mats at different DOX concentrations (100, 50, 20, and 10 μg/mL), and the cells were cultured for 24 h. Finally, the medium in each well was replaced with fresh medium containing 20 μL resazurin solution (0.1 mg/mL in PBS), and the cells were cultured for another 4 h before evaluation by a microplate reader (model Victor3 1420, PerkinElmer). Meanwhile, the morphology of the treated cancer cells was observed by an inverted microscope (Nikon Eclipse TE 2000E).

Comment 5:

It is important to know what the incubation conditions and after how long the release media from the electrospun nanofibers were incubated with cells. It would be worth to study time dependent release media mediated toxicity studies.

Author reply: According to the reviewer’s suggestion, we have provided this information on Page 5 Line 11-18. See below:

“A certain amount of experimental and control samples was weighed, and all samples needed UV sterilization overnight. All the samples were placed in a 6-well cell culture plate, and then about 2 mL of 75% alcohol was added to soak and disinfect for 5 min, while UV irradiation was carried out for sterilization. Then, the alcohol was sucked out and each well was washed repeatedly with PBS for 3 times. We then added an appropriate volume of RPMI 1640 medium into each well (the theoretical concentration of DOX in nanofibers was 200 μg/mL), and put the cell culture plate in a CO2 incubator for 24 h at 37 oC. The immersion medium of each well was collected for subsequent CAL72 cell culture.”

Comment 6:

I believe In vivo experiments should be performed to prove the efficacy of the developed composite nanofibers.

Author reply: This is a good idea. Due to the limitation of conditions, we did not verify its antitumor effect in vivo, but we will carry out relevant animal experimental studies when conditions are available in the near future. Thank you for your great opinion.

Reviewer 2 Report

The manuscript is written well and interesting. All preparation procedures are well described, the protocol of the observations seem comprehensive and considerable.  

-However, there are a some issues that should be clarified or corrected by the authors:

-2.2. In vitro antitumor activity evaluation/ In vitro antitumor efficacy: antitumor is not correct term, it should be antiproliferative activity.

Line 100 -Concentration of ATT/DOX complex, DOX/PLGA or  ATT/DOX/PLGA nanofibrous  has not been mentioned. Also add recent reference for this protocol.

  • There are large numbers of grammar error. Needs to be checked for spelling and grammatical mistakes like misspelled words, errors with punctuation, etc 

-In abstract. Authors must include one sentence about the background and need of this study.

- Authors must describe a clear cut novelty and objective of the present work in abstract.

- Supplementary file fig S3. All the images are look like seem.

- Conclusion: authors should write sentence about the advantages and future application of present work

Author Response

Reviewer #2:

Comments to the Author:

The manuscript is written well and interesting. All preparation procedures are well described, the protocol of the observations seem comprehensive and considerable. 

However, there are a some issues that should be clarified or corrected by the authors:

Comment 1: 2.2. In vitro antitumor activity evaluation/ In vitro antitumor efficacy: antitumor is not correct term, it should be antiproliferative activity.

Author reply: We thank the reviewer for his/her great comments. According to the reviewer’s suggestion, we have changed “antitumor activity” to “antiproliferative activity” in the corresponding text of the revised manuscript.

Comment 2:

Line 100 -Concentration of ATT/DOX complex, DOX/PLGA or ATT/DOX/PLGA nanofibrous has not been mentioned. Also add recent reference for this protocol.

Author reply: According to the reviewer’s suggestion, we have provided the concentrations of ATT/DOX complexes, DOX/PLGA or ATT/DOX/PLGA nanofibrous mats on Page 5 Line 19-22 in the revised manuscript. See below:

CAL72 cells (2 × 103 cells per well) were plated into a 96-well plate and cultured in a humidified incubator overnight. Then, the medium was replaced with fresh medium containing free DOX, and the release medium of the ATT/DOX complex, DOX/PLGA or ATT/DOX/PLGA nanofibrous mats at different DOX concentrations (100, 50, 20, and 10 μg/mL), and the cells were cultured for 24 h.

Comment 3: There are large numbers of grammar error. Needs to be checked for spelling and grammatical mistakes like misspelled words, errors with punctuation, etc.

Author reply: According to the reviewer’s suggestion, we have carefully checked and corrected spelling and grammatical mistakes in the revised manuscript.

Comment 4: In abstract. Authors must include one sentence about the background and need of this study.

Author reply: According to the reviewer’s suggestion, we have added one sentence about the background and need of this study in abstract on Page 2 Line 2-3 in the revised manuscript. See below:

Currently, cancer chemotherapeutic drugs still have the defects of high toxicity and low bioavailability, so it is critical to design novel drug release systems for cancer chemotherapy.

Comment 5: Authors must describe a clear cut novelty and objective of the present work in abstract.

Author reply: We have highlighted the novelty and objective of the present work in the abstract on Page 2 Line 12-15. See also below:

“Owing to the significantly reduced burst release profile and increased mechanical durability of the ATT/DOX/PLGA nanofibers, the designed organic–inorganic hybrid nanofibers may hold great promise as a nanoplatform to encapsulate different drugs for enhanced tumor local therapy applications.”

Comment 6: Supplementary file fig S3. All the images are look like seem.

Author reply: The values of water contact angles of PLGA, ATT/PLGA, and ATT/DOX/PLGA nanofibers were quite similar to each other, so the images look like seem.

Comment 7: Conclusion: authors should write sentence about the advantages and future application of present work.

Author reply: According to the reviewer’s suggestion, we have described the advantages and future application of the present work on Page 14 Line 7-15 in the revised manuscript. See below:

“ATT/DOX-doped PLGA nanofibers can significantly improve the mechanical properties of electrospun PLGA nanofibers, which could be used as a functional material for tissue engineering scaffold for enhanced tumor local therapy applications. In addition, ATT particles and PLGA nanofibers have dual sustained-release effects on drug molecules. In future work, the dispersion of ATT and DOX/ATT in electrospinning solution could be further improved through the surface modification of ATT, so that ATT can maximize the mechanical properties of the fibers. Furthermore, in vivo experiments should be performed to prove the antitumor efficacy of the developed composite nanofibers.”

Reviewer 3 Report

Although the paper is moderately novel, it should be improved according to following lines:

1- Authors should provide the FTIR spectra of nanofibers before and after ATT/DOX loading.

2- Authors should add details of mechanical testing tools and the standard method used at materials and method section. 

3- Authors should add the details of equipment used along with the detailed method for release study at materials and method section.

Author Response

Reviewer #3:

Comments to the Author:

Although the paper is moderately novel, it should be improved according to following lines:

Comment 1:Authors should provide the FTIR spectra of nanofibers before and after ATT/DOX loading.

Author reply: We thank the reviewer for his/her great comments. We have provided the FTIR spectra of ATT, ATT/DOX, and free DOX in Figure 1.

Comment 2: Authors should add details of mechanical testing tools and the standard method used at materials and method section.

Author reply: We have provided these details in Supporting Information. See below:

Characterization techniques

The ATT/DOX/PLGA composite nanofibers were characterized via different techniques. Fourier transform infrared spectroscopy (FTIR) was performed using a Nicolet Nexus 670 FTIR spectrometer over a wavenumber range of 500 to 4000 cm-1 to confirm the loading of DOX onto the ATT particles. The crystalline structures of ATT before and after modifications were analyzed by a Rigaku D/max-2550 PC X-ray diffraction (XRD) system (Rigaku Co., Tokyo, Japan) with a wavelength of 0.154 nm at 40 kV and 200 mA. The scan was performed from 5° to 70°. Lastly, the DOX, ATT, and ATT/DOX solutions were characterized via UV-vis spectroscopy (Perkin Elmer Lambda 25, Waltham, MA) at a wavelength range of 200-900 nm. The morphology of the ATT/DOX nanohybrid was characterized by transmission electron microscopy (TEM, JZM-2100, Japan) at an operating voltage of 200 kV. The morphologies of PLGA, ATT/PLGA, PLGA/DOX, and ATT/DOX/PLGA nanofibers were also observed by SEM (JEOL JSM-5600LV, Tokyo, Japan) at a voltage of 15 kV. The porosity, mechanical properties, and the surface hydrophilicity of the nanofibers were measured according to protocols described in our previous work [1].

Comment 3: Authors should add the details of equipment used along with the detailed method for release study at materials and method section.

Author reply: We also provided these details in the Supporting Information. See below:

In vitro drug release

The release kinetics of DOX was determined by measuring the DOX absorbance at 490 nm using a UV-vis spectrophotometer. The ATT/DOX complex (5 mg) was dispersed in 2 mL of phosphate buffer saline (PBS, pH = 7.4) or sodium acetate–acetic acid buffer solution (pH = 5.4). The dispersed solution was then transferred to a dialysis tube, which was placed in a vial containing 8 mL of the corresponding buffer solution. Similarly, PLGA/DOX and ATT/DOX/PLGA nanofibers with the same DOX concentration were directly placed into different vials containing 10 mL of the corresponding buffer solution. All the samples were incubated in a vapor-bathing constant temperature vibrator with a shaking speed of 90 rpm at 37 oC for a period of 10 days. At each predetermined time point, 1 mL of outerphase solution was removed from each vial for quantitative analysis using UV-vis spectroscopy. An equal volume of fresh corresponding buffer solution was added to the vial.”

Round 2

Reviewer 1 Report

There is no significant improvement of why Attapulgite is better over most widely explored laponite or hydroxyapatite or other electrospun PLGA nanofibers. It is recommended to compare the results of Attapulgite PLGA nanofibers with other electrspun nanofibers to clearly demonstrate the significance of using the Attapulgite in the PLGA electrospun nanofibers. The authors don’t understand how to write materials and methods section. The methods must be clearly mentioned and give details of how the experiment is performed. For example, “Fourier transform infrared spectroscopy (FTIR) was performed using a Nicolet Nexus 670 FTIR spectrometer over a wavenumber range of 500 to 4000 cm-1 to confirm the loading of DOX onto the ATT particles.” There is nothing mentioned about sample preparation. Were the samples directly given to the instrument and asked the instrument to analyze the results? What was the sample concentrations used for analysis? There are many details missing in the materials and methods. How was the data analyzed? What is the statistical significance? What method was used to analyze the statistical significance? Without these fundamental details, it is not advisable to allow the manuscript for publication. The In Vitro release experiment mentions that the volume of the sample used for dialysis is 2 mL, which was placed in the dialysis release medium of 8 mL. In theory, it is advisable to use at lease 10 times more column volume to generate a sink condition. However, using 8 mL volume is not ideal. Please check the previous literature for performing the release experiments. The authors mention the use of UV-Vis spectroscopy for measuring the release of DOX. What was the excitation and emission wavelengths used? Authors were asked to collect the released media from the PLGA nanofibers at different time points, incubate them with the CAL72 cells to investigate time dependent release mediated toxicity. There is no experiment results for the question asked. Another flaw in the experimental section is authors say in the first paragraph that “We then added an appropriate volume of RPMI 1640 medium into each well (the theoretical concentration of DOX in nanofibers was 200 μg/mL), and put the cell culture plate in a CO2 incubator for 24 h at 37 oC” However, in the second paragraph the authors wrote Then, the medium was replaced with fresh medium containing free DOX, and the release medium of the ATT/DOX complex, DOX/PLGA or ATT/DOX/PLGA nanofibrous mats at different DOX concentrations (100, 50, 20, and 10 μg/mL), and the cells were cultured for 24 h.” Were the release samples prepared with theoretical DOX concentrations of 100 μg/mL or 200 μg/mL? Overall, the article needs to be examined in detail and make necessary changes to present the article in publishable form.

Author Response

Reviewer #1:

 Comment 1:

There is no significant improvement of why Attapulgite is better over most widely explored laponite or hydroxyapatite or other electrospun PLGA nanofibers. It is recommended to compare the results of Attapulgite PLGA nanofibers with other electrspun nanofibers to clearly demonstrate the significance of using the Attapulgite in the PLGA electrospun nanofibers.

Author reply: We thank the reviewer for his/her great comments. According to the reviewer’s suggestion, we have compared the characteristics of ATT-doped electrospun nanofibers with reported HNT-, LAP-, and n-HA-based electrospun nanofibers. Due to the high porosity, small dimension, high aspect ratio and relatively uniform morphology of ATT, the ATT-doped nanofibers show a much higher drug loading efficiency than the reported HNT-, LAP-, and n-HA-based electrospun nanofibers, which is beneficial for antitumor applications. To make it clearer, we have added one sentence to describe it on Line 63-66 in the revised manuscript. See also below:

“Due to the high porosity, small dimension, high aspect ratio and relatively uniform morphology of ATT, the ATT-doped nanofibers show a much higher drug loading efficiency than the reported HNT-, LAP-, and n-HA-based electrospun nanofibers, which is beneficial for antitumor applications [49-51].”

Comment 2:

The authors don’t understand how to write materials and methods section. The methods must be clearly mentioned and give details of how the experiment is performed. For example, “Fourier transform infrared spectroscopy (FTIR) was performed using a Nicolet Nexus 670 FTIR spectrometer over a wavenumber range of 500 to 4000 cm-1 to confirm the loading of DOX onto the ATT particles.” There is nothing mentioned about sample preparation. Were the samples directly given to the instrument and asked the instrument to analyze the results? What was the sample concentrations used for analysis? There are many details missing in the materials and methods. How was the data analyzed? What is the statistical significance? What method was used to analyze the statistical significance? Without these fundamental details, it is not advisable to allow the manuscript for publication.

Author reply: We thank the reviewer for his/her great comments. To address the review’s concerns, we have added the details of sample preparation for FTIR, TEM, and SEM, data analysis for SEM, and statistical analysis in the revised Supporting Information. See also below:

“For Fourier transform infrared spectroscopy (FTIR), we used potassium bromide pressed-disk technique to prepared samples. That is to say, we took 2-3 mg of sample (free ATT, DOX, or the ATT/DOX particles) and 200-300 mg of dry KBr powder, mixed them in agate mortar, grinded them fully, and pressed the sample to disk before measurements according to the manufacturer’s protocol.”

“Aqueous dilute suspensions of ATT and ATT/DOX (5 μL) were dropped onto carbon-coated copper grid and air dried before TEM measurements. For the PLGA nanofibers and ATT/DOX/PLGA composite nanofibers, the fiber sample was directly electrospun onto the carbon-coated copper grid and vacuum dried before TEM imaging.”

“All samples were sputter coated with gold films with a thickness of 10 nm before SEM observation.”

“The diameters of the electrospun fibers were analyzed using ImageJ 1.53k software (http://rsb.info.nih.gov/ij/download.html, National Institutes of Health, USA). At least 200 nanofibers at different images were analyzed for each sample to get the diameter distribution histogram.”

“Statistical analysis

One way ANOVA statistical analysis was performed to compare the antitumor efficacy of DOX in different formulations with PBS as a negative control. A value of 0.05 was selected as the significance level, and the data were indicated with (*) for p< 0.05, (**) for p < 0.01, and (***) for p < 0.001, respectively.”

Comment 3:

The In Vitro release experiment mentions that the volume of the sample used for dialysis is 2 mL, which was placed in the dialysis release medium of 8 mL. In theory, it is advisable to use at lease 10 times more column volume to generate a sink condition. However, using 8 mL volume is not ideal. Please check the previous literature for performing the release experiments.

Author reply: We are sorry to make this mistake, we have corrected the description of the in vitro drug release in the revised Supporting Information. See also below:

“In vitro drug release

The release kinetics of DOX was determined by measuring the DOX absorbance at 490 nm using a UV-vis spectrophotometer. The ATT/DOX complex (5 mg) was dispersed in 1 mL of phosphate buffer saline (PBS, pH = 7.4) or sodium acetate–acetic acid buffer solution (pH = 5.4). The dispersion was then transferred to a dialysis tube, which was placed in a vial containing 10 mL of the corresponding buffer solution. Similarly, PLGA/DOX and ATT/DOX/PLGA nanofibers with the same DOX concentration were directly placed into different vials containing 10 mL of the corresponding buffer solution. All the samples were incubated in a vapor-bathing constant temperature vibrator with a shaking speed of 90 rpm at 37 oC for a period of 10 days. At each predetermined time point, 1 mL of outerphase solution was removed from each vial for quantitative analysis using UV-vis spectroscopy. An equal volume of fresh corresponding buffer solution was replenished to the vial to keep the outer phase volume constant.”

Comment 4:

The authors mention the use of UV-Vis spectroscopy for measuring the release of DOX. What was the excitation and emission wavelengths used?

Author reply: For UV-vis spectroscopy, there is no need to set up parameters of excitation and emission wavelengths. Please keep in mind, we used UV-vis spectroscopy, instead of fluorescence spectroscopy to quantify the DOX concentration.

Comment 5:

Authors were asked to collect the released media from the PLGA nanofibers at different time points, incubate them with the CAL72 cells to investigate time dependent release mediated toxicity. There is no experiment results for the question asked?

Author reply: We thank the reviewer for his/her great comments. According to the literature (Zheng et al. Polym. Chem., 2013, 4, 933–941), incubation of the drug-loaded nanofibers within the cell culture medium for 24 h at 37 oC is sufficient to enable the drug release for in vitro anticancer activity assay. Therefore, we just performed the anticancer activity assay using the released media from the PLGA-based nanofibers at just one time point of 24 h (Figures 7 and 8). Since our hypothesis is readily proven just one time point, we decided not to test the time-dependent release mediated toxicity as suggested by the reviewer. In any case, we appreciate the reviewer’s valuable point, which should be concerned in our future development of fiber-based drug delivery systems.

Comment 6:

Another flaw in the experimental section is authors say in the first paragraph that “We then added an appropriate volume of RPMI 1640 medium into each well (the theoretical concentration of DOX in nanofibers was 200 μg/mL), and put the cell culture plate in a CO2 incubator for 24 h at 37 oC” However, in the second paragraph the authors wrote” Then, the medium was replaced with fresh medium containing free DOX, and the release medium of the ATT/DOX complex, DOX/PLGA or ATT/DOX/PLGA nanofibrous mats at different DOX concentrations (100, 50, 20, and 10 μg/mL), and the cells were cultured for 24 h.” Were the release samples prepared with theoretical DOX concentrations of 100 μg/mL or 200 μg/mL?

Author reply: We thank the reviewer for his/her great comments. To address the review’s concerns, we have re-written the experiment of in vitro antiproliferative activity evaluation on Line 108-126 in the revised manuscript. The release medium of each nanofiber with theoretical DOX concentrations is 200 μg/mL.

“The cell suspension with a cell density of 1×105 cells/mL was prepared, and 20 μL was added to a 96-well plate, so that the plating density of CAL 72 cells was 2 × 103 cells/well. Then, 100, 50, 20, or 10 μL fresh medium, free DOX solution ([DOX] = 200 μg/mL), ATT/DOX solution ([DOX] = 200 μg/mL), ATT particle solution with equivalent concentration of the ATT/DOX complexes, and release medium of the nanofibers of DOX/PLGA ([DOX] = 200 μg/mL according to theoretic DOX loading within the fibers), PLGA with equivalent DOX concentration of the release medium of DOX/PLGA, ATT/PLGA with equivalent DOX concentration of the release medium of DOX/PLGA, and ATT/DOX/PLGA ([DOX] = 200 μg/mL according to theoretic DOX loading within the fibers). The total volume was added to 200 μL/well using fresh medium. Each sample and concentration gradient were six parallel, and the samples were shaken well and then cultured in an incubator. Due to the initial concentration of DOX is 200 μg/mL, the concentration gradients after dilution are 100, 50, 20 and 10 μg/mL, respectively. After 1 day of culture at 37 oC and 5% CO2, the original medium was discarded, and 180 μL fresh medium and 20 μL resazurin solution was added into each well (0.1 mg/mL dissolved in PBS and filtered for sterilization). Four hours after re-culture, cell viability was detected by a microplate reader (model Victor3 1420, PerkinElmer). Meanwhile, the morphology of the treated cancer cells was observed by an inverted optical microscope (Nikon Eclipse TE 2000E).”

Reviewer 3 Report

It is acceptable now.

Author Response

Comments and Suggestions for Authors

It is acceptable now.

Author reply: Thank you for your great support. Accordingly, no alteration is required.

Round 3

Reviewer 1 Report

The article can be accepted for publication